# Future prevalence of type 2 diabetes—A comparative analysis of chronic disease projection methods

**Dina Voeltz**[1,2]*, **Thaddäus Tönnies**[3], **Ralph Brinks**[2,3,4], **Annika Hoyer**[1]

**1** Biostatistics and Medical Biometry, Medical School OWL, Bielefeld University, Bielefeld, Germany,
**2** Department of Statistics, Ludwig-Maximilians-University Munich, München, Germany, **3** Institute for Biometrics and Epidemiology, German Diabetes Center, Leibniz Institute for Diabetes Research at Heinrich-Heine-University Duesseldorf, Duesseldorf, Germany, **4** Chair for Medical Biometry and Epidemiology, Faculty of Health/School of Medicine, Witten/Herdecke University, Witten, Germany

* dina.voeltz@uni-bielefeld.de

**Data Availability Statement:** The authors confirm that the data supporting the findings of this study are available within the articles of Tamayo et al. [23], Schmidt et al. [2] and their respective supplementary materials. Other data about the

## Abstract

### Background

Accurate projections of the future number of people with chronic diseases are necessary for effective resource allocation and health care planning in response to changes in disease burden.

### Aim

To introduce and compare different projection methods to estimate the number of people with diagnosed type 2 diabetes (T2D) in Germany in 2040.

### Methods

We compare three methods to project the number of males with T2D in Germany in 2040. Method 1) simply combines the sex- and age-specific prevalence of T2D in 2010 with future population distributions projected by the German Federal Statistical Office (FSO). Methods 2) and 3) additionally account for the incidence of T2D and mortality rates using partial differential equations (PDEs). Method 2) models the prevalence of T2D employing a scalar PDE which incorporates incidence and mortality rates. Subsequently, the estimated prevalence is applied to the population projection of the FSO. Method 3) uses a two-dimensional system of PDEs and estimates future case numbers directly while future mortality of people with and without T2D is modelled independently from the projection of the FSO.

### Results

Method 1) projects 3.6 million male people with diagnosed T2D in Germany in 2040. Compared to 2.8 million males in 2010, this equals an increase by 29%. Methods 2) and 3) project 5.9 million (+104% compared to 2010) and 6.0 million (+116%) male T2D patients, respectively.

population projections used in this study are openly available from the FSO [20]. We provide our complete R-code and underlying data sets for reproducing the analysis in the supporting information.

**Funding:** The author(s) received no specific funding for this work.

**Competing interests:** The authors have declared that no competing interests exist.

## Conclusions

The results of the three methods differ substantially. It appears that ignoring temporal trends in incidence and mortality may result in misleading projections of the future number of people with chronic diseases. Hence, it is essential to include these rates as is done by method 2) and 3).

## Introduction

The increasing proportion of people suffering from chronic diseases is peculiarly worrying as these conditions constrain activities of daily living, necessitate ongoing medical attention and hence, are associated with considerably higher costs and healthcare expenses [1]. Disease management activities have been shown to reduce the risk of acute complications and premature mortality caused by chronic diseases [2]. Though, effective responses require accurate estimates of current and future chronic disease burden to tailor health care planning and resource allocation [3]. Worldwide, diabetes mellitus is one of the most frequent chronic diseases, and thus, is a disease with high public health relevance [1, 2, 4]. Current type 2 diabetes (T2D) prevalence is estimated with 7.4% among men and 7.0% among women in Germany aged 40 years or older [4, 5]. Besides severe late complications, diabetes mellitus leads to significantly higher mortality and is associated with 1.5 to 4.4 times higher health-care costs compared to people without diabetes [1].

Projection models are powerful tools to estimate future case numbers of a disease in order to inform decision-makers and cost-bearers in the health care system. Consequently, further developing and spreading the knowledge about accurate projection methods is essential to counteract the ever-worsening disease situation. However, different models may vary in their outcomes and closeness to reality [6]. Nonetheless, to our best knowledge, there is no scientific work that systematically compares these different projections methods for the context of chronic diseases. Therefore, the aim of this work is to fill this gap and to introduce, describe, and critically discuss each of the methods individually and in comparison.

In general, projection methods are limited by the availability of epidemiological and demographic data. Consequently, it is common to project and compare several scenarios that reflect on possible future trends in these areas [3, 4, 6–8]. Besides, the choice of the method can considerably affect projection results. In the context of chronic diseases, there are several methodological approaches that have been advocated for case number projections. For example, most reports are based on a 'status quo approach' [6] which relies on a simple application of the current prevalence to population projections. This procedure has been used for instance in the contexts of pulmonology [9], Parkinson's disease [10] and diabetes [11] among others. Probably due to its simplicity, this method is most popular in projection contexts. However, approaches that aim to incorporate for example underlying disease-specific transition rates, i.e., incidence and mortality, seem more appropriate as they better capture the complex nature of chronic diseases [12]. Therefore, some studies rely on multistate models that incorporate and relate disease-specific transition rates. In this regard, multistate models are widely used in infectious and chronic disease epidemiology [13–15]. For example, Milan and Fetzer [6], Brinks et al. [8, 16] and Waldeyer et al. [3] used time-discrete Markov models in the context of dementia, lupus and diabetes. Another approach is used by Carstensen et al. [7] who used a Poisson regression to model disease-specific transition rates of diabetes. Thereof, they extrapolate the future trends of the rates by extending the trends observed in the past. A relatively

novel approach is reported by Tönnies et al. [4], who use a partial differential equation to project future diabetes prevalence in Germany. There remains, however, considerable debate about the methodological approach.

In the present article, we give an overview of possible methods and underlying data used for future case number projections in the context of chronic conditions. For this purpose, we will examine three projection methods in more detail. We discuss how to employ each of the approaches in a practical application to project sex- and age-specific case numbers of people with diagnosed T2D for Germany between 2010 and 2040.

## Methods

In the following section, we describe different models for projecting chronic disease case numbers. All methods were implemented using the free software R, v.4.1.0 (The R Foundation for Statistical Computing).

We focus on three different approaches to project chronic disease case numbers. Previous studies have mostly used a very simple approach, commonly referred to as status quo method [6]. Due to its popularity, we include this procedure in our work, speaking of it as method 1). This approach solely relies on the age- and sex-specific prevalence from a base year, which is then applied to population projections. Other epidemiological factors, such as the incidence, are only incorporated implicitly in the prevalence. Thus, this method ignores the fact that prevalence is a consequence of incidence and mortality. Hence, it might be too simplistic to accurately mirror reality. The alternative methods 2) and 3) rely on demographic components as well as on various disease-specific information on prevalence, incidence, and mortality rates as input factors. Method 2) is aligned with the work of Tönnies et al. [4], who takes advantages of the theory of multistate models in chronic disease epidemiology and an associated partial differential equation (PDE). With method 3) we present a novel projection method, which consists of a two-dimensional system of PDEs. The theoretical background for the PDEs used with method 2) and 3) originates in the illness-death model (IDM) as depicted in Fig 1. The IDM is a multistate model that represents continuous-time stochastic processes. Thereby, it allows individuals to move between a finite number of states [17, 18]. The classical IDM consists of three states, i.e., "healthy" (number of healthy people aged $a$ at time $t$ $H(t, a)$) with regards to the disease of interest, the disease state "ill" (number of ill people $I(t, a)$) and the death state, i.e., "dead" ($D$). It is assumed that at birth all individuals start in the healthy state. From there on, they can either be diagnosed with a chronic disease like T2D and then die at some point in time, or they can transition directly to death state (without contracting diabetes). The arrows indicate the transition rates between the states which depend on age and time. Since diabetes is a chronic condition, we assume that there is no remission from the chronic

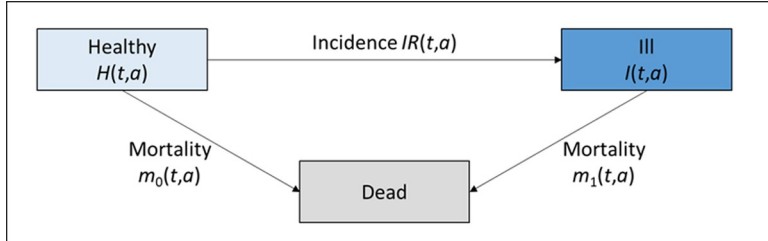

**Fig 1. Illness-death model.** All people in a population are in one of the three states: Healthy, Diseased, or Dead. It is assumed that at birth, all people start in the healthy state. Depending on time $t$ and age $a$ of each respective person, they will then transition to another state which is described by the $IR$, $m_0$, and $m_1$.

condition back to the healthy state. The transition rates are given by the incidence rate $IR(t, a)$ the mortality of the non-diseased $m_0(t, a)$ and the mortality of diseased people $m_1(t, a)$, which are all sex-specific functions of calendar time $t$ and age $a$. In epidemiological contexts, calendar time $t$ is also denoted by period.

The following sections describe each of the three methods as sketched in Fig 2 and the required input data in more detail. Aggregated data about prevalence and incidence of the chronic condition of interest, as well as on the mortality of the general population and the excess mortality are sufficient, i.e., none of the methods require individual subject data.

## Data

The starting point for the projections are disease-specific as well as demographic input factors. The required demographic information essentially comprises the expected age and sex-specific population distribution in Germany for each year that is to be included in the projection. The disease-specific input factors include the age- and sex-specific prevalence, incidence rate, and mortality rates of the diseased and non-diseased along with information on the age- and sex-specific excess mortality (mortality rate ratio, MRR).

We used published claims data on prevalence and incidence of diagnosed diabetes in 2009 and 2010 from 65 million people insured by the German public health insurance funds [19]. Diabetes was determined by the International Classification of Diseases-10 codes E10–E14. Overall, approximately 10.1% had any type of diabetes mellitus (excluding gestational diabetes), while 7.3% were diagnosed with T2D in 2010.

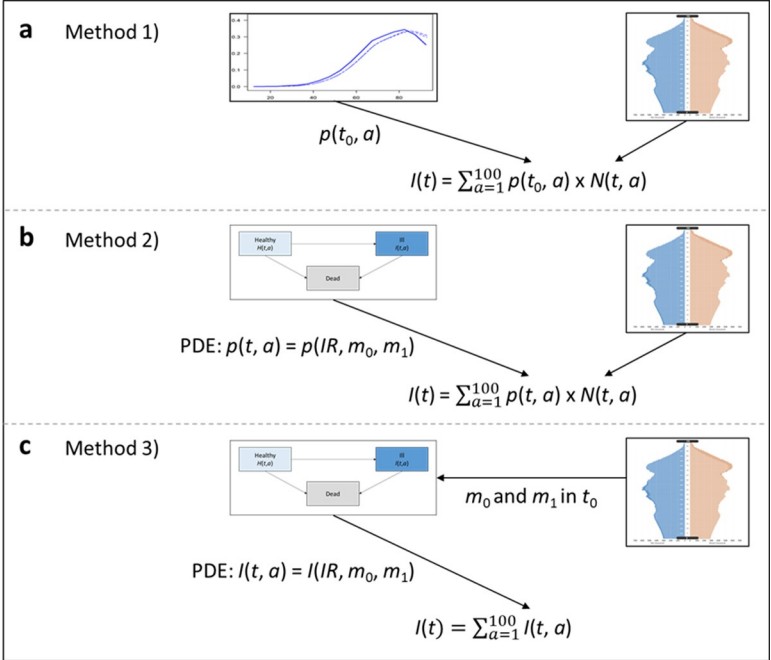

**Fig 2. Overview of methods for future disease case projection.** Illustrated in Fig 2A, method 1) uses the age-specific prevalence in base year $t_0$ which is applied to the population projections. Fig 2B depicts method 2). Using the theoretical background of the IDM, the age-specific prevalence for each year is derived by solving the PDE which requires input on the transition rates, namely the IR, $m_0$ and $m_1$. Method 3), as sketched in Fig 2C, calibrates $m_0$ and $m_1$ in the base year $t_0$ from the population projections. $IR$, $m_0$ and $m_1$ are inputs for the PDE which directly returns age-specific T2D case numbers for each year. Summing over all ages yields the total number of T2D cases for each year from 2010 to 2040 for each method.

Concerning the population distribution, we used data from the German Federal Statistical Office (FSO) [20] that regularly issues updated projections of future population numbers in Germany. These projections include different scenarios regarding expected birth rate, life expectancy and migration. All projection scenarios contain sex-specific results for all ages from 0 to 100 years for the time horizon between 2010 and 2040. In our main analysis, we focus on one selected variant, namely B1L2M1. This variant assumes a birth rate (B) of 1.4 children per woman, a life expectancy (L) at birth in 2040 of 84.4 years for men and a long-term net migration (M) of 147,000 people. The motivation for this was twofold: First, we decided to consider more realistic variants instead of including more extreme options. Second, using variant B1L2M1 aligns with Tönnies et al. [4] who used the same variant for their projection. Nonetheless, in the supporting information we provide results for other variants as well, thereby showing that using different population projections of the FSO on the prevalence projection seems negligible.

Further, we obtain input values for the mortality rate of the general German population between 2010 and 2040 from the population projections of the German FSO [20]. Distinct information on the mortality rate of the healthy, i.e., non-diseased with regard to T2D, and for the mortality rate of the diseased would be of interest, but unfortunately, the mortality of the healthy population in Germany with respect to T2D is unknown. To cope with this lack of data, we substitute the missing mortality rates for method 2) with a mathematically equivalent expression based on the general mortality and the MRR as an alternative epidemiological measure. For method 3), we show how to calculate the mortality rates of the diseased and the healthy population.

The MRR is based on a similar, nationally representative dataset from 2014 reported by Schmidt et al. [2]. Unfortunately, the MRR is not differentiated by diabetes type. However, since T1D is frequent at ages younger than 20 and T2D is more common among older ages, the MRR is mostly driven by deaths among the latter. Moreover, most diabetes cases are attributable to T2D. Therefore, we use the MRR estimates provided by Schmidt et al. [2] as an approximation of the T2D-related MRR in Germany.

Further, reliable information on the temporal trend of the diabetes-specific MRR is relatively restricted in Germany. Following the work of Brinks et al. [8] and Tönnies et al. [4], we therefore refer to trends in the sex- and age-specific MRR observed in Denmark [21]. The motivation to do so is twofold. Firstly, it has been shown that for countries that are comparable in terms of their disease burden and health care systems, such as Denmark and Germany, the MRR settles in a similar range [22]. More precisely, a 2% decrease in the MRR per year is reported for Denmark [21]. Secondly, the same approach is used by Tönnies et al. [4], who also assume a decrease of 2% per year in the MRR as observed in other countries.

## Method 1—The simple approach

Method 1) combines diabetes prevalence with population projections as depicted in Fig 2A. Specifically, to project the number of T2D cases until 2040, we multiplied German age- and sex-specific population projections provided by the FSO with age- and sex-specific T2D prevalence in 2010 from Tamayo et al. [23]. The latter is assumed to remain constant. The age- and sex-specific prevalence in 2010, $p(t, a)$ is determined by:

$$p(2010, a) = \frac{I(2010, a)}{H(2010, a) + I(2010, a)} \tag{1}$$

with $I(2010, a)$ being the number of people diagnosed with T2D in 2010 and $H(2010, a)$ the number of people without T2D in 2010. The underlying mathematical relation to calculate the

age- and sex-specific future number of cases $I(t, a)$ is then given by:

$$I(t, a) = N(t, a) \times p(2010, a) \tag{2}$$

where $N(t, a)$ denotes the age-, sex- and time-dependent total population number and $p(2010, a)$ the age-specific prevalence in the year 2010.

## Method 2—The two-step multistate model

Method 2) refines the first method. To this end, we use mathematical relations to incorporate the relation between prevalence, incidence, and mortality, as well as temporal changes in the incidence and mortality [24]. More precisely, with method 2) we firstly model the temporal change in prevalence of T2D employing a PDE [25]. Secondly, we compute future T2D case numbers by multiplying the age- and sex-specific projected prevalence with the respective projected population size (Fig 2B).

Brinks et al. [26] showed that the change in prevalence can be modelled by a PDE in case information about mortality, incidence and prevalence at a specific time point is given. The PDE is given by

$$\partial p = (1 - p) \times [IR - p \times (m_1 - m_0)] \tag{3}$$

In other words, method 2) relies on the relation between prevalence $p(t, a)$, the incidence rate $IR(t, a)$ and the mortality rates $m_0(t, a)$ and $m_1(t, a)$. Unfortunately, $m_0(t, a)$ is unknown for most diseases. Therefore, we use $m(t, a)$ the mortality of the general population, and the age- and sex-specific $MRR(t, a)$ which denotes the ratio of the two mortality rates, i.e. $\frac{m_1(t,a)}{m_0(t,a)}$. The general mortality is defined as

$$m = p \times m_1 + (1 - p)m_0 \tag{4}$$

The PDE can then be rewritten as

$$\partial p = (1 - p) \times \left[ IR - \frac{p \times (MRR - 1) \times m}{p \times (MRR - 1) + 1} \right] \tag{5}$$

For the present application to the context of T2D, this allows us to project the change in prevalence at a certain age and time even without data about $m_0(t, a)$. Accordingly, we integrate the PDE shown in Eq (5). Doing so, we use nationally representative input values for the projected general mortality of the German population as provided by the FSO and the observed age-specific prevalence of T2D in year 2010 among all individuals within the German statutory health insurance as initial prevalence. Recall that the equation is particularly favourable since the PDE allows for incorporating the complex interplay of the IR and MRR, as well as it is able to reflect on temporal trends of these rates. By application and integration of the PDE we derive the estimated age- and sex-specific prevalence $\hat{p}(t, a)$ for all following years considered in the time horizon of the projection, i.e., here from 2010 to 2040. In a final step, this estimated prevalence $\hat{p}(t, a)$ is multiplied with the population projections of the FSO [20] as follows

$$I = N \times \hat{p} \tag{6}$$

This yields case numbers of T2D in Germany $I(t, a)$ for each year between 2010 and 2040 for all ages ranging from 18 to 100 years. All ages below 18 years were considered negligible due to minimal prevalence at these ages [4, 23].

## Method 3—The two-dimensional PDE model

Similar to method 2), the third method builds upon the IDM. Though, method 3) features two PDEs as sketched in Fig 2C to directly express changes in the age- and sex-specific numbers of diabetes cases, i.e., $I(t, a)$, and the changes in the healthy population $H(t, a)$. We can show that these two figures $I(t, a)$ and $H(t, a)$ are interrelated with the transition rates $IR(t, a)$, $m_0(t, a)$ and $m_1(t, a)$ via the following equations

$$\partial H = -(IR + m_0) \times H \tag{7}$$

$$\partial I = IR \times H - m_1 \times I \tag{8}$$

$\partial H(t, a)$ indicates age-, sex- and time-specific changes in the number of people without diabetes, while $\partial I(t, a)$ represents changes in the number of cases. As $m_0(t, a)$ is unknown for Germany, we use the T2D prevalence from 2010, the general mortality $m(t, a)$ and the $MRR(2014, a)$ to derive values for $m_0(t, a)$ and $m_1(t, a)$ [21, 22, 27]. Precisely, $m_0(t, a)$ and $m_1(t, a)$ are calculated by

$$m_0 = \frac{m}{(1 + p \times (MRR - 1))} \tag{9}$$

$$m_1 = m_0 \times MRR \tag{10}$$

In contrast to method 2), we only need data on the general mortality and MRR in 2010 for method 3) which is then used to determine $m_0(t, a)$ and $m_1(t, a)$. Besides, we derive information on the development of the future birth rate of the population projections. However, using only information on the annual number of new-borns is little critical for our projection period, as the projection ends in 2040. That means, someone born in 2020 will then be 20 years old and will thus not belong to the diabetes risk group. That means we do not need further information from population projections that are based on historical data and thus subject to retrograde developments as provided by the FSO [20]. This is favourable as these population projections do not explicitly take specific diseases into account. In contrast, using method 3) and a two-dimensional system of PDEs allows us to reflect on short-term changes especially with regards to relevant disease-specific alterations. As outcome, the PDEs directly describe the age- and sex-specific number of people with and without T2D for each year between 2010 and 2040.

## Results

Based on the age-specific prevalence in 2010, we projected the age-specific case numbers for T2D in the German male population between 2010 and 2040 using the above-mentioned approaches. Remember that we assume a constant age-specific prevalence with method 1), i.e., we apply the prevalence from 2010 to our population projections. We further assumed that also the IR remains constant over time for method 2) and 3) in the main analysis. However, we consider temporal trends in the MRR for method 2) and 3) which affects, amongst others, the prevalence. Since for Germany, information on long-term trends in the latter are limited, we use speculative time-related developments. Current evidence undermines that the mortality rate among people with T2D is likely to decrease faster than among healthy people due to progresses in medical care [21]. Therefore, it seems plausible to assume a reduction in the MRR [28]. For simplicity, we restrict the main analysis to a baseline scenario for method 2) and 3) which thus comprises a constant IR and a decrease in the MRR of 2% per year. We reflect on

**Fig 3. T2D case projection results.** Projected male T2D cases in Germany for variant B1L2M1 of the population projections of the German FSO between 2010 and 2040. The population projection variant is based on constant birth rates, increases in life expectancy and a net migration of 147 000 people. Method 1) assumes constant age-specific prevalence, method 2) and method 3) build on constant incidence rate and 2%-decrease in the MRR.

rather likely, as well as on relatively extreme scenarios for the MRR and IR in the sensitivity analyses discussed in the supporting information.

Fig 3 depicts the number of projected male T2D cases in Germany between 2010 and 2040 using the three different approaches. All the proposed methods suggest that the number of people with diabetes continues to grow in Germany. However, the projected number of male T2D cases differs substantially across the three methods. It is clearly visible that over time the difference between the projected values from each of the three methods enlarges. Since only the population composition changes over time, method 1) shows only slightly increasing future case numbers. In contrast, methods 2) and 3) also take into account the incidence and mortality, which influence the prevalence and number of cases. This, in addition to the demographic aging, is above all the reason for the higher number of projected T2D cases for these methods compared to method 1).

Table 1 displays the projected T2D case numbers among men in Germany in millions and the relative changes from 2010 compared to 2040 in percent. With method 1), we find an

**Table 1. T2D case projection results (in million).**

| Method | 2010 | 2020 | 2030 | 2040 | Absolute difference | Relative difference |
|---|---|---|---|---|---|---|
| 1) | 2.81 | 3.21 | 3.47 | 3.63 | 0.82 | 29.09% |
| 2) | 2.93 | 4.36 | 5.33 | 5.97 | 3.04 | 103.60% |
| 3) | 2.81 | 4.16 | 5.25 | 6.08 | 3.27 | 116.34% |

Results are based on variant B1L2M1 of the German FSO (absolute (in Million) and relative (in percent) difference with regards to 2010 vs. 2040).

increase of male T2D case numbers by 0.8 million. This equals a total of almost 3.6 million T2D cases among males in Germany in 2040 (+29%). When presuming that the MRR decreases by 2% per year while the IR is assumed to remain constant, the number of cases is projected to increase by about 3.0 million (+104%) and 3.3 million (+116%) for method 2) and method 3), respectively. Apparently, the results show an obvious gap between method 1) and the other two methods. Interestingly, the difference between the projected case numbers of the latter methods is minor: During the entire time horizon of the projection until 2035, the future number of T2D cases for method 2) is slightly higher than the number projected by method 3). Thereafter, method 3) projects a higher number of male T2D cases in Germany.

S1 and S2 Figs in S2 File refer to the age-specific prevalence among males in Germany from 2010 to 2040 projected using the different methods. We find apparent discrepancies in the age-specific prevalence for the methods. Compared to the constant prevalence assumed by method 1), the prevalence computed with methods 2) and 3) is expected to increase considerably, particularly for people older than 60 years. Additionally, the peak prevalence is presumed to shift towards older age groups. We provide further details in the supporting information.

## Sensitivity analyses

We performed several sensitivity analyses to assess inaccuracies and uncertainty in our results which we briefly discuss in this section. First, we assessed uncertainty due to unknown future trends in the model parameters. In order to inspect uncertainty that could arise due to sampling error of the input values, we perform a Monte Carlo simulation for all three methods. To evaluate the latter, i.e., uncertainty due to future trends, we analyse 14 different scenarios of declining or rising MRR and IR with methods 2) and 3) following the example of Tönnies et al. [4]. More detailed information on results and computation is given in the supplementary information.

General future epidemiological trends for the age-specific IR or the MRR are heterogeneous and precise information are not available for Germany [4, 5]. Therefore, we considered 14 different scenarios of declining or rising MRR and IR and examined the impact on future case numbers using method 2) and 3). We find considerably large deviations in the number of projected T2D cases: With the most extreme scenarios we project a decrease of approx. 0.3 million (-11%; stable MRR & IR -5%) or an increase of 6.3 million (216%; MRR -2% & IR +3%) male T2D cases between 2010 and 2040 for Germany (S3 Fig in S2 File). This is an essential finding, as it underlines how impactful disease-specific factors are in projection contexts. This result is strongly in favour of method 2) and 3), as these approaches incorporate the rates underlying the prevalence in the mathematical model.

In order to assess uncertainty in our results, we calculated 95% confidence intervals using a Monte Carlo simulation as shown in S4 Fig in S2 File. The sampling error of the input values led to deviations in the future number of cases by approximately 7.2% for method 2) and by about 3.5% for method 3). Hence, the Monte Carlo simulation shows that the uncertainty due to sampling error in the input values seems to be of limited relevance. We report further information about the calculation and the estimation results in the supporting information.

## Discussion

The aim of this article is to assess and advance the usage and performance of different projection methods in chronic disease modelling. While there is a lack thereof in previous publications, we compared the underlying assumptions, mathematical details, strengths, and limitations of three distinct projection methods and demonstrated their usage. Precisely, in this article, we discussed three methods to project age- and sex-specific case numbers of a

chronic condition from age- and sex-specific prevalence, incidence, and mortality data. Further, we illustrated each method in a practical application in the context of T2D among males in Germany for the time period from 2010 to 2040. We found considerable differences between the results of the three methods.

Generally, all three methods can be easily adapted to other chronic diseases or countries. Additionally, for Germany, data about the future population number is available until 2060, i.e., the projection of T2D case numbers can be extended easily to the far-off future. Nonetheless, a projection is subject to some unforeseen changes in the economic, political, or disease-specific environment amongst others. Generally, the reliability of projections decreases over time. Therefore, instead of delving further into the future and thereby decreasing certainty, we restricted the projection until 2040.

In many previous projections on chronic diseases, age- and sex-specific prevalence from a base year is transferred to population projections as in method 1) [6, 9–11]. Population projections required for this purpose, generated using the so-called cohort component method [6, 29], are widely used and publicly available in Germany. For the calculation, the birth cohorts are hypothetically extrapolated from a base year under assumptions on the future development of fertility, mortality, and net migration. But just as the future population depends on several components, the development of a disease is also influenced by various disease-specific aspects [6]. However, the latter are ignored in method 1). Without a doubt, method 1) is the least reliable of the three discussed approaches. Assuming a constant prevalence and not reflecting on changes in disease-specific factors substantially limit the reliability of this method's results. Unfortunately, in some contexts, data on the latter may be rarely available. If that is the case, method 1) could at least provide a short-term indication of future case numbers. Furthermore, short-term projections could reflect a second context in which method 1) may be applicable. A projection is always subject to some unforeseen changes in the economic, political, or disease-specific environment amongst others. Consequently, when keeping the overall time-horizon to a minimum, changes in relevant factors such as the mortality rate ratio or the incidence rate may be minimal and therefore less impactful, too. Nonetheless, method 1) should be used with caution and whenever possible, method 2) or 3) should be the preferred options. Methods 2) and 3) include disease-specific information on mortality, prevalence, and incidence rates as input factors in addition to the demographic components mentioned above. Thus, as opposed to method 1), the PDEs used in method 2) and 3) account for temporal trends in the prevailing epidemiological situation. To do so, aggregated data are sufficient. Method 2) incorporates the MRR and IR, as opposed to solely using the prevalence as is done by method 1). Though, method 2) is, as method 1), still based on projections of the population and the general mortality in Germany. The future estimates of the German FSO do not consider potential epidemiological influences and simply extrapolate future values based on long-term trends of the past [20]. Contrarily, using method 3), we can take short-term fluctuations in the mortality into account. The results of method 3) are independent from projections of the German FSO beyond the year used to estimate $m_0$ and $m_1$. Instead, this method purely relies on input values on the age- and sex-specific prevalence, the MRR and the general mortality for a base year. The latter are used to estimate the mortality of the non-diseased and the mortality of diseased people. Furthermore, instead of calculating the prevalence in each year for each age group, method 3) directly returns future changes in absolute case and population numbers.

Besides, some health actions or interventions may not directly alter the prevalence of a disease but still have far-reaching consequences with regards to a particular disease situation [28, 30, 31]. Consequently, other health indicators such as life expectancy, years of life lost, or years of productivity lost may provide a more appropriate insight to anticipate the potential impact of such interventions. Nonetheless, in most cases, a disease's prevalence is a necessary input

factor for the computation of other health indicators. Accordingly, using the illness–death model and the corresponding PDE in a first step to estimate a future prevalence allows to derive other health characteristics of a population associated with T2D in Germany in 2020 and 2040 from an individual and population perspective in a second step [28, 31]. In fact, this reflects another drawback of method 1). In contrast to method 2) and 3), the approach assumes a stable prevalence, and consequently, this does not allow to easily infer the future development of other epidemiological indicators.

Lastly, different future scenarios that anticipate potential impacts of alternative scenarios of prevention and intervention or disease-dynamics can be computed and compared using method 2) or 3). The comparison of methods 2) and 3) with method 1) reveals the great impact of trends in incidence and mortality on future disease burden. Our findings confirm the suggestion that ignoring temporal trends in incidence and mortality provokes an underestimation of the actual number of cases [4]. Consequently, a future course can be better, if not optimal, approximated using method 2) and 3).

As a further remark, we are convinced that our work provides a good basis for future research. Our study discusses the large importance of choosing a suitable projection method which is supported by a practical application to the context of T2D in Germany. Since our approach to use a PDE that describes the IDM is not the only one that has been proposed and published [31, 32], it would be interesting to extend the comparison and focus on different approaches that are based on the IDM. Since this would have been beyond the scope of this article, this investigation could prove desirable for future work.

## Limitations

A primary limitation for method 2) and 3) is that they need assumptions about the temporal changes in the IR and MRR. Unfortunately, information about future trends in Germany in the context of T2D is scarce. Therefore, we made assumptions about future trends of the IR and MRR that may be oversimplified. Further, it is noteworthy that our population projection methods are trend-based, hence, they may be less accurate for instance in periods of sudden and fast changes in the incidence. However, also the intensity, frequency and pace of changes have implications on epidemiologically relevant measures such as the prevalence and the number of cases. As discussed, variant projections, which are feasible with method 2) and 3), can provide additional information by illustrating potential alternative scenarios of future trends. Aside from that, rapid changes in chronic disease incidences are rare and the general impact of high-frequency distortions in future trends is often limited by the inertia in population change.

Another limitation of all three methods arises with a potential violation of the assumption that the prevalence in migrants is equal to the prevalence observed among German residents. Though, the T2D prevalence in people migrating from and to Germany is unknown. However, other examples show that this issue is minor and overall epidemiological measures are only negligibly affected [4, 25]. Though, one should keep in mind that it may remain an issue e.g., for small populations.

Another weakness of the methods arises with the ignored effect of potential covariates and their development. Examples for relevant covariates in the diabetes context might be a change of diagnostic criteria for T2D, the distribution of body weight, the impact of nutritional behaviour, or the presence of co-morbidities. In an epidemiological context, it may be essential to consider such covariates since they likely modify the transition rates between the states in the IDM. Although this is not done in this work, it is possible to account for the impact of possible covariates such as interventions and risk-factors in the PDEs [33].

## Conclusion

Forecasts of the growing non-communicable disease burden will be key to guide future health-care policies. With this work, we compare three projection methods and demonstrated how to apply each of them to quantify future T2D case numbers in Germany until 2040. The three methods uniformly confirm that there is a substantial increase in the number of males diagnosed with T2D ranging from 0.8 million (+29%) to 3.3 million (+116%) additional cases in Germany in 2040. Assessing the strengths and limitations of three different methods may help researchers to better apply statistical methods for projecting future case numbers of chronic diseases. We suggest that future projections should move away from blunt prevalence extrapolation and instead, employ methods that are based on theory from the IDM.

## Supporting information

**S1 File. R code.**
(PDF)

**S2 File.**
(DOCX)

## Author Contributions

**Conceptualization:** Annika Hoyer.

**Methodology:** Dina Voeltz, Ralph Brinks, Annika Hoyer.

**Writing – original draft:** Dina Voeltz.

**Writing – review & editing:** Dina Voeltz, Thaddäus Tönnies, Ralph Brinks, Annika Hoyer.

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
