## [Decision Letter · Decision Letter 0]

18 Jan 2022

PONE-D-21-38063Future prevalence of type 2 diabetes – a comparative analysis of chronic disease projection methodsPLOS ONE

Dear Dr. Voeltz,

Thank you for submitting your manuscript to PLOS ONE. After careful consideration, we feel that it has merit but does not fully meet PLOS ONE’s publication criteria as it currently stands. Therefore, we invite you to submit a revised version of the manuscript that addresses the points raised during the review process.

We look forward to receiving your revised manuscript.

Kind regards,

Bedreddine Ainseba, PhD

Academic Editor

PLOS ONE

Journal Requirements:

Reviewers' comments:

Reviewer's Responses to Questions

**Comments to the Author**

1. Is the manuscript technically sound, and do the data support the conclusions?

Reviewer #1: Yes

Reviewer #2: Yes

Reviewer #3: Yes

2. Has the statistical analysis been performed appropriately and rigorously? 

Reviewer #1: I Don't Know

Reviewer #2: Yes

Reviewer #3: Yes

3. Have the authors made all data underlying the findings in their manuscript fully available?

Reviewer #1: Yes

Reviewer #2: Yes

Reviewer #3: Yes

4. Is the manuscript presented in an intelligible fashion and written in standard English?

Reviewer #1: Yes

Reviewer #2: Yes

Reviewer #3: Yes

5. Review Comments to the Author

Reviewer #1: Referee Report on the manuscript "Future prevalence of type 2 diabetes a comparative analysis of chronic disease projection methods" submitted by Dina Voeltz.

The author studies 3 differents models in order to project the number of males with T2D in Germany in 2040. The first method is a model based on a simple formula to compute the age- and sex-specific prevalence p(t,a) and then to predict the futur number of cases I(t,a) including an age-,sex- and time-dependent population projections provided by the German FSO.I would like to ask the authors to rewrite this last reference by FSO in References section. The second method is a two-step multistate model and this method refines the above first method by incorporating a relation between prevalence p(t,a), incidence IR(t,a) and mortality given by a ratio. The third method is different and based on a two-dimensional simple PDE model builds upon a multistate model that allow individuals to move between states. In the simulation the authors perform a sensitivity analysis and i recommand the authors to bring together the sections Sensitivity analysis at page 11 and Sensitivity analyses at page 14 of the paper. I believe that the results stated in the paper are correct so that i can recommend the publication of this paper.

Without having time to manipulate the R-codes, i believe that the results stated in the paper are correct so that i can recommend the publication of this paper.

Reviewer #2: The aim of the paper is clearly motivated, as authors want to introduce and compare different methods for the estimation of the number of people with diagnosed type 2 diabetes (T2D) in Germany in 2040.

Method 1 is a commonly and simply approach assuming a constant age-specific prevalence, whereas Methods 2 and 3 are taking into account the incidence of T2D and mortality rates by using partial differential equations.

An important gap is clearly noticeable between method 1 and methods 2 and 3 for which the future number of T2D case is higher. It appears to be essential to include temporal trends in incidence rate and mortality rate ratio for a better projection.

COMMENTS

1) In equation 6), please define what exactly is the notation p hat.

2) Line 257, I guess that number 28 refers to the citation [28]. Please use the correct notation for citing this reference.

3) On the description of “S1 Fig. Comparing T2D prevalence projections”, line 471, the projected age-specific prevalence is supposed to be in 2015, 2020, 2030 and 2040. However, on the specific figure, the year 2010 appears instead of 2015.

Same remark for the figure and the text in the "supporting information 1" document. Make sure that the title and the figure are corrected related to the attached explanations.

4) On this same "S1 Fig. Comparing T2D prevalence projections", how can you explain the difference of method 2 with respect to Methods 1 and 3 for the 2010 results? Indeed, it seems that methods 1 and 3 fit pretty well together in 2010, unlike method 2. It should be discussed, in comparison to the other years where methods 2 and 3 better fit together unlike method 1.

5) On "S2 Fig. Projected prevalence across future population variants", there is no explanation on the choice of the different variants studied. Only B1L2M1 is described at the beginning of the article. A quick mention of the other variants studied (e.g. why have chosen these ones especially?) in the figure could be wise for the convenience of the reader non familiar with this kind of chronic disease.

Reviewer #3: This manuscript deals with a comparative analysis of different methods of projection of the prevalence of chronic diseases.

In this manuscript, it is a comparison of 3 already published methods so I will not go into the details of the methods but just propose some additional discussion points.

The first approaches, even if it is used in a "natural" way by epidemiologists, is undoubtedly the least "reliable" if one does not have homogeneity of many factors in time (incidence, composition of the population,...) especially if one projects far from the observed data. It's still a good thing to have compared the two other to this one because it is often chosen. But may be it could have been made even clearer, in the discussion part, in which context the first method can be used.

I would add to the advantages of not choosing it the fact of not being able to easily calculate other epidemiological indicators. A small criticism of this paper is that it does not mention the possibility, and the interest, of being able to calculate other indicators (life expectancy, healthy or sick, lifetime risk of a chronic disease, average age of onset of the disease, expectation of the age of onset of the disease, ...) and to focus only on prevalence. Indeed, we can see in other papers (see Wanneveich et al 2018 for example) that sometimes health actions may not decrease the prevalence (or even increase it) simply by increasing life expectancy but also by increasing the age of onset of the disease... Indeed, in addition, the possibility of modelling interventions or changes in the prevalence of risk factors is also a major point to take into account.

Another point, possibly to be discussed or added as information, is that the approach presented here with an IDM is not the only one that has been proposed and published using an IDM (example Jacqmin-Gadda et al 2013). And therefore the results of the simulations presented are only relative to one particular approach using an IDM. It would have been interesting (but this is certainly beyond the scope of this article) to be able to compare different approach using IDM between them.

Jacqmin-Gadda, H., Alperovitch, A., Montlahuc, C., Commenges, D., Leffondre, K., Dufouil, C., ... & Joly, P. (2013). 20-Year prevalence projections for dementia and impact of preventive policy about risk factors. European journal of epidemiology, 28(6), 493-502.

Wanneveich, M., Jacqmin-Gadda, H., Dartigues, J. F., & Joly, P. (2018). Impact of intervention targeting risk factors on chronic disease burden. Statistical methods in medical research, 27(2), 414-427

6. PLOS authors have the option to publish the peer review history of their article (what does this mean?). If published, this will include your full peer review and any attached files.

Reviewer #1: **Yes: **Solym MANOU-ABI

Reviewer #2: No

Reviewer #3: No

---

## [Author Response · Author response to Decision Letter 0]

2 Feb 2022

General comment for all reviewers and the associate editor

We thank the associate editor and the reviewers for evaluating our manuscript and sending us their constructive feedback. After a further critical view on our paper, we completely agree with them and modified the manuscript according to their comments. Below is our point-to-point response to each comment raised by the associate editor and the reviewers. We hope that we satisfyingly addressed the reviewers’ suggestions, and that the manuscript is now suitable for publication.

Reviewers' comments:

Reviewer 1

1) The author studies 3 differents models in order to project the number of males with T2D in Germany in 2040. The first method is a model based on a simple formula to compute the age- and sex-specific prevalence p(t,a) and then to predict the futur number of cases I(t,a) including an age-,sex- and time-dependent population projections provided by the German FSO.I would like to ask the authors to rewrite this last reference by FSO in References section. 

Answer:

We rewrote the reference of the FSO population projections. It is now referencing the English website and not further the webpage of the “Statistisches Bundesamt”.

2) The second method is a two-step multistate model and this method refines the above first method by incorporating a relation between prevalence p(t,a), incidence IR(t,a) and mortality given by a ratio. The third method is different and based on a two-dimensional simple PDE model builds upon a multistate model that allow individuals to move between states. In the simulation the authors perform a sensitivity analysis and i recommand the authors to bring together the sections Sensitivity analysis at page 11 and Sensitivity analyses at page 14 of the paper. I believe that the results stated in the paper are correct so that i can recommend the publication of this paper. Without having time to manipulate the R-codes, i believe that the results stated in the paper are correct so that i can recommend the publication of this paper.

Answer:

We thank the reviewer for the suggestion. We decided to follow the recommendation and now jointly discuss the Sensitivity analysis in a final subsection of the Result section. 

 

Reviewer 2

The aim of the paper is clearly motivated, as authors want to introduce and compare different methods for the estimation of the number of people with diagnosed type 2 diabetes (T2D) in Germany in 2040. Method 1 is a commonly and simply approach assuming a constant age-specific prevalence, whereas Methods 2 and 3 are taking into account the incidence of T2D and mortality rates by using partial differential equations. An important gap is clearly noticeable between method 1 and methods 2 and 3 for which the future number of T2D case is higher. It appears to be essential to include temporal trends in incidence rate and mortality rate ratio for a better projection.

1) In equation 6), please define what exactly is the notation p hat.

Answer: 

We thank the reviewer for the hint. We modified the paragraph in the method section which should now more clearly define the notation p ^: 

Method: 

Accordingly, we integrate the PDE shown in equation (5). Doing so, we use nationally representative input values for the projected general mortality of the German population as provided by the FSO and the observed age-specific prevalence of T2D in year 2010 among all individuals within the German statutory health insurance as initial prevalence. Recall that the equation is particularly favourable since the PDE allows for incorporating the complex interplay of the IR and MRR, as well as it is able to reflect on temporal trends of these rates. By application and integration of the PDE we derive the estimated age- and sex-specific prevalence p ^(t, a) for all following years considered in the time horizon of the projection, i.e., here from 2010 to 2040. In a final step, this estimated prevalence p ^(t, a) is multiplied with the population projections of the FSO [20] as follows

 I= N× p ^ (6)

2) Line 257, I guess that number 28 refers to the citation [28]. Please use the correct notation for citing this reference.

Answer: 

We thank the reviewer for that hint and corrected the notation.

3) On the description of “S1 Fig. Comparing T2D prevalence projections”, line 471, the projected age-specific prevalence is supposed to be in 2015, 2020, 2030 and 2040. However, on the specific figure, the year 2010 appears instead of 2015. Same remark for the figure and the text in the "supporting information 1" document. Make sure that the title and the figure are corrected related to the attached explanations.

Answer: 

We are grateful for that hint and corrected the mistakes.

4) On this same "S1 Fig. Comparing T2D prevalence projections", how can you explain the difference of method 2 with respect to Methods 1 and 3 for the 2010 results? Indeed, it seems that methods 1 and 3 fit pretty well together in 2010, unlike method 2. It should be discussed, in comparison to the other years where methods 2 and 3 better fit together unlike method 1.

Answer: 

We thank the reviewer for the hint and checked S1 Fig. Indeed, there was a mistake in the plot which referred to 2015 for method 2) and to 2010 for method 1) and 3). We corrected it accordingly and inserted the appropriate figure in the text. S1 Fig now shows that all three methods initially return the same age-specific prevalence of T2D among males in Germany 2010 before beginning with the projection for the time period until 2040. 

5) On "S2 Fig. Projected prevalence across future population variants", there is no explanation on the choice of the different variants studied. Only B1L2M1 is described at the beginning of the article. A quick mention of the other variants studied (e.g. why have chosen these ones especially?) in the figure could be wise for the convenience of the reader non familiar with this kind of chronic disease.

Answer: 

We agree with the reviewer that the motivation for using variant B1L2M1 in the main analysis and the choice of different variants in the supporting material is somewhat short. We clarified the choice of the used variant in the main analysis and modified the paragraph in the supporting information in the following way: 

Data: 

In our main analysis, we focus on one selected variant, namely B1L2M1. This variant assumes a birth rate (B) of 1.4 children per woman, a life expectancy (L) at birth in 2040 of 84.4 years for men and a long-term net migration (M) of 147,000 people. The motivation for this was twofold: First, we decided to consider more realistic variants instead of including more extreme options. Second, using variant B1L2M1 aligns with Tönnies et al. [4] who used the same variant for their projection. Nonetheless, in the supporting information we provide results for other variants as well, thereby showing that using different population projections of the FSO on the prevalence projection seems negligible.

Supporting information: 

In our main analysis, we focus on one selected variant, namely B1L2M1. This variant assumes a birth rate of 1.4 children per woman (B1), a life expectancy at birth in 2040 of 84.4 years for men (L2) and a long-term net migration of 147,000 people (M1). The motivation for this was twofold: First, we decided to consider more realistic variants instead of including more extreme options. Second, using variant B1L2M1 aligns with Tönnies et al. [4] who used the same variant for their projection of T2D in Germany. Though, for a more profound view, S2 Fig provide results for variants B1L2M1, B1L2M2, B2L2M1 and B2L2M2 to assess the impact changes in the birth rate to 1.6 children per woman (B2), a life expectancy at birth in 2040 of 82.5 years for men (L1) and a long-term net migration of 221,000. The set of variants was chosen for the same two reasons as stated above.

 

Reviewer 3

1) This manuscript deals with a comparative analysis of different methods of projection of the prevalence of chronic diseases. In this manuscript, it is a comparison of 3 already published methods so I will not go into the details of the methods but just propose some additional discussion points. The first approaches, even if it is used in a "natural" way by epidemiologists, is undoubtedly the least "reliable" if one does not have homogeneity of many factors in time (incidence, composition of the population,...) especially if one projects far from the observed data. It's still a good thing to have compared the two other to this one because it is often chosen. But may be it could have been made even clearer, in the discussion part, in which context the first method can be used.

Answer: 

We thank the reviewer and apologize for this lack of clarity. As it is stated in the discussion of the paper, we think that method 1) is indeed the least reliable one. Assuming a constant prevalence and not reflecting on changes in disease-specific factors is a major limitation of the approach. Unfortunately, in some contexts, data on the latter may rarely be available. If that is the case, method 1) could at least provide a short-term indication of future case numbers. Nonetheless and without a doubt, method 1) should be used with caution and whenever possible, method 2) or 3) should be the preferred options. Furthermore, short-term projections could reflect a second context in which method 1) may be applicable. A projection is always subject to some unforeseen changes in the economic, political, or disease-specific environment amongst others. Consequently, when keeping the overall time-horizon to a minimum, changes in relevant factors such as the mortality rate ratio or the incidence rate will in most cases be minimal, too. 

For better clarity and a more detailed understanding we added the following paragraph to our discussion:

Discussion:

Without a doubt, method 1) is the least reliable of the three discussed approaches. Assuming a constant prevalence and not reflecting on changes in disease-specific factors substantially limit the reliability of this method’s results. Unfortunately, in some contexts, data on the latter may be rarely available. If that is the case, method 1) could at least provide a short-term indication of future case numbers. Furthermore, short-term projections could reflect a second context in which method 1) may be applicable. A projection is always subject to some unforeseen changes in the economic, political, or disease-specific environment amongst others. Consequently, when keeping the overall time-horizon to a minimum, changes in relevant factors such as the mortality rate ratio or the incidence rate may be minimal and therefore less impactful, too. Nonetheless, method 1) should be used with caution and whenever possible, method 2) or 3) should be the preferred options.

2) I would add to the advantages of not choosing it the fact of not being able to easily calculate other epidemiological indicators. A small criticism of this paper is that it does not mention the possibility, and the interest, of being able to calculate other indicators (life expectancy, healthy or sick, lifetime risk of a chronic disease, average age of onset of the disease, expectation of the age of onset of the disease, ...) and to focus only on prevalence. Indeed, we can see in other papers (see Wanneveich et al 2018 for example) that sometimes health actions may not decrease the prevalence (or even increase it) simply by increasing life expectancy but also by increasing the age of onset of the disease... Indeed, in addition, the possibility of modelling interventions or changes in the prevalence of risk factors is also a major point to take into account.

Answer: 

We agree with the reviewer that this is a fair point to mention. The aim of our work was to focus on a statistical view for comparing the different projection methods. Consequently, rather than immersing into the epidemiological context and potentially distracting readers with a load of possibly interesting health indicators, we limited our elaborations to the prevalence only. Nonetheless, we appreciate the hint of the reviewer that the discussion of other health indicators may be of interest. Therefore, we modified the discussion in the following way and include the mentioned reference to Wanneveich:

Discussion: 

Besides, some health actions or interventions may not directly alter the prevalence of a disease but still have far-reaching consequences with regards to a particular disease situation [28,30,31]. Consequently, other health indicators such as life expectancy, years of life lost, or years of productivity lost may provide a more appropriate insight to anticipate the potential impact of such interventions. Nonetheless, in most cases, a disease's prevalence is a necessary input factor for the computation of other health indicators. Accordingly, using the illness–death model and the corresponding PDE in a first step to estimate a future prevalence allows to derive other health characteristics of a population associated with T2D in Germany in 2020 and 2040 from an individual and population perspective in a second step [28,31]. In fact, this reflects another drawback of method 1). In contrast to method 2) and 3), the approach assumes a stable prevalence, and consequently, this does not allow to easily infer the future development of other epidemiological indicators.

3) Another point, possibly to be discussed or added as information, is that the approach presented here with an IDM is not the only one that has been proposed and published using an IDM (example Jacqmin-Gadda et al 2013). And therefore the results of the simulations presented are only relative to one particular approach using an IDM. It would have been interesting (but this is certainly beyond the scope of this article) to be able to compare different approach using IDM between them.

Jacqmin-Gadda, H., Alperovitch, A., Montlahuc, C., Commenges, D., Leffondre, K., Dufouil, C., ... & Joly, P. (2013). 20-Year prevalence projections for dementia and impact of preventive policy about risk factors. European journal of epidemiology, 28(6), 493-502.

Wanneveich, M., Jacqmin-Gadda, H., Dartigues, J. F., & Joly, P. (2018). Impact of intervention targeting risk factors on chronic disease burden. Statistical methods in medical research, 27(2), 414-427

Answer: 

We thank the reviewer for the recommendation and for pointing to these two additional articles. We added them in our manuscript to the corresponding new discussion part: 

Discussion:

As a further remark, we are convinced that our work provides a good basis for future research. Our study discusses the large importance of choosing a suitable projection method which is supported by a practical application to the context of T2D in Germany. Since our approach to use a PDE that describes the IDM is not the only one that has been proposed and published [31,32], it would be interesting to extend the comparison and focus on different approaches that are based on the IDM. Since this would have been beyond the scope of this article, this investigation could prove desirable for future work.

---

## [Decision Letter · Decision Letter 1]

16 Feb 2022

Future prevalence of type 2 diabetes – a comparative analysis of chronic disease projection methods

PONE-D-21-38063R1

Dear Dr. Dina Voeltz

We’re pleased to inform you that your manuscript has been judged scientifically suitable for publication and will be formally accepted for publication once it meets all outstanding technical requirements.

Kind regards,

Bedreddine Ainseba, PhD

Academic Editor

PLOS ONE

Reviewers' comments:

Reviewer's Responses to Questions

**Comments to the Author**

1. If the authors have adequately addressed your comments raised in a previous round of review and you feel that this manuscript is now acceptable for publication, you may indicate that here to bypass the “Comments to the Author” section, enter your conflict of interest statement in the “Confidential to Editor” section, and submit your "Accept" recommendation.

Reviewer #2: All comments have been addressed

Reviewer #3: (No Response)

2. Is the manuscript technically sound, and do the data support the conclusions?

Reviewer #2: (No Response)

Reviewer #3: (No Response)

3. Has the statistical analysis been performed appropriately and rigorously? 

Reviewer #2: (No Response)

Reviewer #3: (No Response)

4. Have the authors made all data underlying the findings in their manuscript fully available?

Reviewer #2: (No Response)

Reviewer #3: (No Response)

5. Is the manuscript presented in an intelligible fashion and written in standard English?

Reviewer #2: (No Response)

Reviewer #3: (No Response)

6. Review Comments to the Author

Reviewer #2: (No Response)

Reviewer #3: (No Response)

7. PLOS authors have the option to publish the peer review history of their article (what does this mean?). If published, this will include your full peer review and any attached files.

Reviewer #2: No

Reviewer #3: No

---

## [Editor Report · Acceptance letter]

18 Feb 2022

PONE-D-21-38063R1 

Future prevalence of type 2 diabetes – a comparative analysis of chronic disease projection methods 

Dear Dr. Voeltz:

I'm pleased to inform you that your manuscript has been deemed suitable for publication in PLOS ONE. Congratulations! Your manuscript is now with our production department. 

Kind regards, 

on behalf of

Dr. Bedreddine Ainseba 

Academic Editor

PLOS ONE